# Reporting in clinical studies on platelet-rich plasma therapy among all medical specialties: A systematic review of Level I and II studies

**Jaron Nazaroff[1], Sarah Oyadomari[1], Nolan Brown[1], Dean Wang[1,2] ***

**1** University of California Irvine School of Medicine, Irvine, CA, United States of America, **2** Department of Orthopaedic Surgery, University of California Irvine Health, Orange, CA, United States of America

* deanwangmd@gmail.com

## Abstract

**Data Availability Statement:** All relevant data are within the paper and its Supporting Information files.

**Funding:** The author(s) received no specific funding for this work.

### Background

The clinical practice of platelet-rich plasma (PRP) therapy has grown significantly in recent years in multiple medical specialties. However, comparisons of PRP studies across medical fields remain challenging because of inconsistent reporting of protocols and characterization of the PRP being administered. The purpose of this systematic review was to determine the quantity of level I/II studies within each medical specialty and compare the level of study reporting across medical fields.

### Methods

The Cochrane Database, PubMed, and EMBASE databases were queried for level I/II clinical studies on PRP injections across all medical specialties. From these studies, data including condition treated, PRP processing and characterization, delivery, control group, and assessed outcomes were collected.

### Results

A total of 132 studies met the inclusion and exclusion criteria and involved 28 different conditions across 8 specialties (cardiothoracic surgery, cosmetic, dermatology, musculoskeletal (MSK), neurology, oral maxillofacial surgery, ophthalmology, and plastic surgery). Studies on PRP for MSK injuries made up the majority of the studies (74%), with knee osteoarthritis and tendinopathy being most commonly studied. Of the 132 studies, only 44 (33%) characterized the composition of PRP used, and only 23 (17%) reported the leukocyte component. MSK studies were more likely to use patient-reported outcome measures to assess outcomes, while studies from other specialties were more likely to use clinician- or imaging-based objective outcomes. Overall, 61% of the studies found PRP to be favorable over control treatment, with no difference in favorable reporting between MSK and other medical specialties.

**Competing interests:** The authors have declared that no competing interests exist.

## Conclusions

The majority of level I/II clinical studies investigating PRP therapy across all medical specialties have been conducted for MSK injuries with knee osteoarthritis and tendinopathy being the most commonly studied conditions. Inconsistent reporting of PRP composition exists among all studies in medicine. Rigorous reporting in human clinical studies across all medical specialties is crucial for evaluating the effects of PRP and moving towards disease-specific and individualized treatment.

## Introduction

The use of platelet-rich plasma (PRP) to treat a multitude of medical conditions has greatly increased over the past decade. As a strategy to deliver a higher concentration of growth factors and cytokines that initiate and regulate tissue healing, PRP therapy has been utilized for a wide range of orthopaedic injuries, including tendinopathies, osteoarthritis, and muscle injuries [1–3]. Recently, PRP has also been increasingly used for the treatment of cosmetic conditions, including hair restoration, breast augmentation, scar treatment, and dermatologic conditions [4–6]. Other reported applications of PRP therapy have included nerve regeneration, periodontal therapies, wound healing, and augmentation of surgical repairs [7–9].

Despite the widespread clinical practice of PRP in all areas of medicine, there remains uncertainty and skepticism among the medical community regarding its efficacy. Much of this skepticism can be attributed to the unawareness of the quantity and quality of evidence investigating PRP treatment, particularly across medical specialties. The practice of evidence-based medicine utilizes the strongest quality of evidence to make informed decisions on the care of individual patients. Although many randomized controlled trials investigating PRP have been conducted for musculoskeletal (MSK) conditions [1,3,10,11], the number of high-quality studies on PRP treatment from other medical specialties compared to orthopaedics, sports medicine, and other MSK fields is unknown. Furthermore, there remain deficiencies in the level of reporting in these studies, particularly regarding the processing and composition of PRP. This has led to calls within orthopaedics for minimal reporting standards in order to allow for reproducibility and comparison across studies [12–15]. Whether the level of reporting is similarly inconsistent within studies from other medical fields is unknown. Detailed reporting in clinical trials for PRP across all medical fields would be beneficial for identifying the key components of PRP and efficiently translating PRP therapy into clinically meaningful treatment.

The purpose of this systematic review was to review the current PRP literature across all medical specialties and determine 1) the quantity of level I and II studies within each medical specialty based on indication, and 2) the level of reporting in these studies with regards to PRP processing, composition, activation, delivery, and outcome assessment. Due to the majority of these studies being from the orthopaedic literature, comparisons in the level of reporting between MSK studies and those from other medical fields were performed.

## Materials and methods

### Article identification and selection process

A literature search was conducted in June 2019 to identify articles pertaining to PRP therapy according to Preferred Reporting Items for Systematic Review and Meta-Analysis (PRISMA) guidelines (Fig 1) [16]. The PubMed (including MEDLINE), Cochrane, and EMBASE

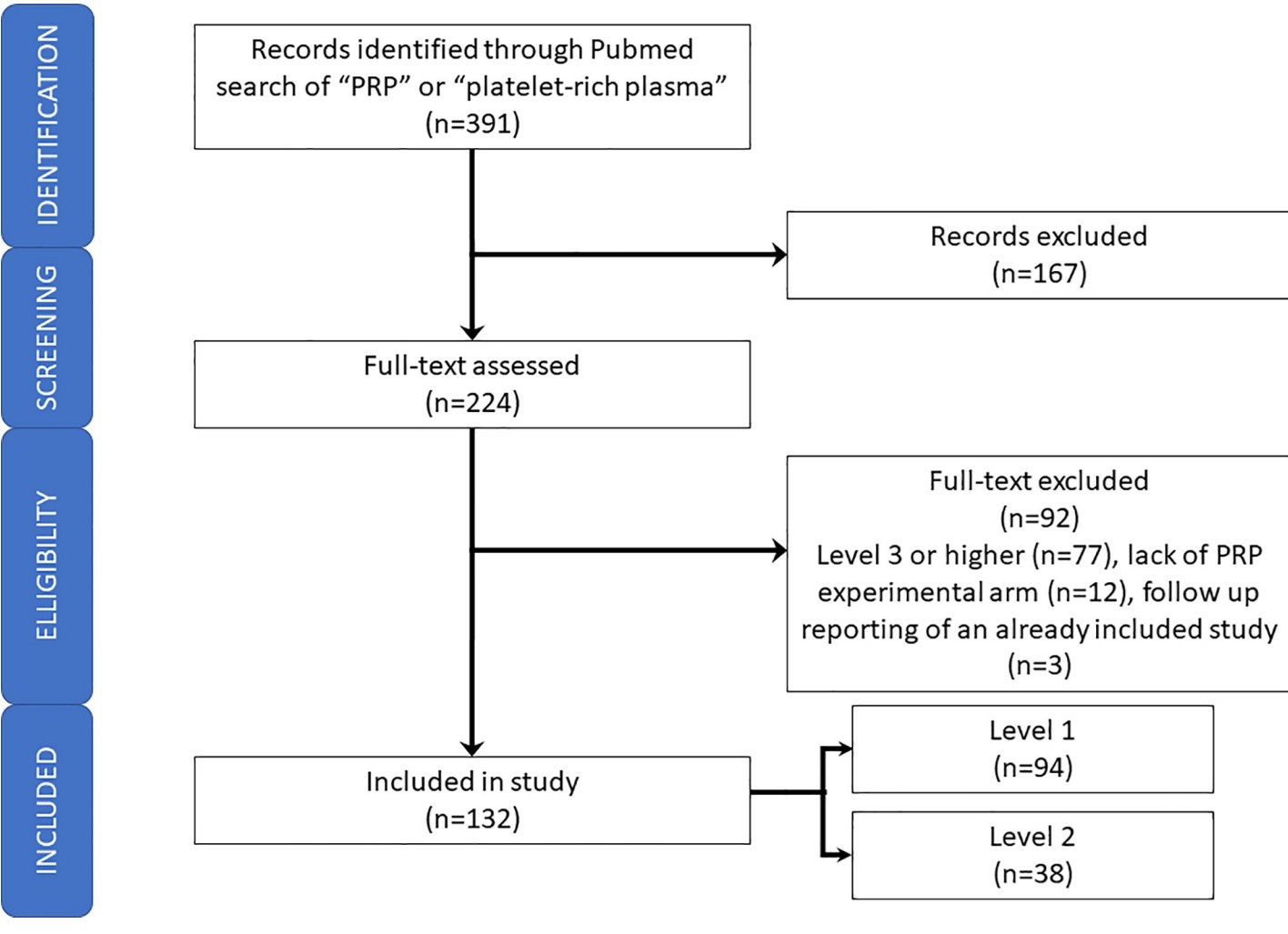

**Fig 1. PRISMA flowchart.**

databases were queried using the following search terms: "platelet-rich plasma," "platelet rich plasma," and "PRP." Inclusion criteria consisted of studies investigating PRP treatment in human clinical trials, comparative level I and level II studies defined by JBJS Journal's Levels of Evidence [17] from all medical specialties, and articles in the English language. Level III-V studies, animal studies, non-comparative, meeting abstracts, book chapters, and systematic reviews and meta-analyses were excluded. After removal of duplicates, article abstracts were first reviewed to identify studies that were consistent with the inclusion and exclusion criteria. The full texts of these articles were then further assessed to determine which studies were eligible for this review. Reference lists of key articles were analyzed for studies to be included in this review. The literature search was performed by two reviewers and the selected articles were reviewed by the senior author.

## Data collection and statistical analysis

A full list of the analyzed studies can be found in S1 Appendix. From the eligible studies, the following data were collected: level of evidence according to JBJS Journal's Levels of Evidence [17], journal name, condition treated, frequency and dosage of PRP administered, time of

outcome assessment, composition of PRP (or the commercial system used), activation state of PRP and activating agent, control and other comparison groups, objective and subjective outcomes measured, and overall conclusion on efficacy or favorability of treatment. Extracted data was independently evaluated by the authors. Studies were grouped into one of the following categories based on the condition being treated: cardiothoracic surgery, cosmetic, dermatology, musculoskeletal (MSK), neurology, oral maxillofacial surgery, ophthalmology, and plastic surgery. Data on the reporting of PRP characteristics focused on the platelet and leukocyte composition. The overall conclusion on PRP efficacy or favorability was determined by the authors' conclusion statement, typically based on statistically significant improvements seen in the outcomes of the PRP treatment group compared to control groups. Included studies were analyzed using the Cochrane Risk of Bias Tool and the Methodological Index for Non-Randomized Studies to asses for bias [18,19]. The senior author reviewed bias assessment scores. Due to the overwhelming number of MSK studies compared to studies from all other specialties, proportional data between studies from MSK and all other specialties was compared using Fisher's exact test.

## Results

### Number of studies

After removal of duplicates, 391 records were examined. One hundred sixty-seven studies were removed after title screening, and 92 studies were removed after full-text examination. Among the studies excluded included those with levels of evidence III or higher (n = 77), lack of a PRP experimental arm (n = 12), and follow-up reporting of an already included study (n = 3). A total of 132 studies met the inclusion and exclusion criteria and were analyzed for this study (Fig 1). Of these studies, 94 (71%) were level I studies, and 38 (29%) were level II studies (S1 Appendix).

Among the analyzed studies, there were 28 different conditions across eight medical fields (Table 1). Studies investigating PRP treatment for MSK conditions comprised 74% of all studies. Tendinopathy (n = 29) and osteoarthritis (n = 28) were the two most commonly studied conditions. MSK studies were 76% level 1 evidence while 57% of all other studies were level 1 evidence (p<0.05). Cosmetic studies comprised 14% (n = 19) of all studies, and 53% of these were level I evidence.

### Reporting of PRP composition and administration

Among all studies, 44 studies (33%) provided details on PRP processing or characteristics of PRP being studied. Within specialty, 37% of MSK studies reported composition details, and 23% of studies from other specialties reported composition details (p = 0.15). Only 23 of the 44 studies also reported details on the leukocyte concentration (Table 2). No studies from the fields of cardiothoracic surgery, ophthalmology, oral maxillofacial surgery, and plastic surgery provided details on platelet or leukocyte counts. A total of 57 studies (43%) indicated that the PRP was activated prior to administration. Calcium was used as the activator in 80% of these studies (Table 3). Among MSK studies, 36% used an activator, whereas 63% of studies from other specialties used an activator (p<0.01). Regarding PRP administration, 84% of studies reported details on quantity, volume, and dosing interval of PRP injections. The quantity of PRP treatments administered ranged from 1 to 9, and the total volume of PRP administered in a single setting ranged from 0.1 to 22 mL. The treatment window ranged from a single injection to serial doses over the span of one year.

**Table 1. Number of Level I and Level II studies investigating platelet-rich plasma treatment organized by medical specialty.**

| Specialty | Condition | Level I | Level II |
|---|---|---|---|
| **Cardiothoracic Surgery (1)** | Blood Loss | 1 | 0 |
| **Cosmetic (19)** | Alopecia | 5 | 1 |
| | Fat graft | 0 | 1 |
| | Hair Regrowth | 2 | 0 |
| | Hyperpigmentation | 0 | 1 |
| | Scars/Stretch marks | 3 | 6 |
| **Dermatology (3)** | Psoriasis | 0 | 1 |
| | Vitiligo | 1 | 1 |
| **Musculoskeletal (97)** | ACL Reconstruction | 1 | 0 |
| | Arthroplasty/Arthroscopy | 0 | 2 |
| | Back Pain | 1 | 0 |
| | Carpal Tunnel | 1 | 0 |
| | Disk Degeneration | 1 | 0 |
| | Fracture | 4 | 0 |
| | Lumbar Facet Syndrome | 1 | 0 |
| | Meniscus Repair | 2 | 0 |
| | Muscle Injury | 4 | 1 |
| | Osteoarthritis | 21 | 7 |
| | Plantar Fasciitis | 7 | 5 |
| | Rotator Cuff Repair | 6 | 2 |
| | Sprain | 1 | 1 |
| | Tendinopathy | 24 | 5 |
| **Neurology (3)** | Carpal Tunnel | 1 | 1 |
| | Neuropathy | 1 | 0 |
| **Ophthalmology (1)** | Retinitis Pigmentosa | 0 | 1 |
| **Oral Maxillofacial Surgery (6)** | Temporomandibular Joint Disorder | 6 | 0 |
| **Plastic Surgery (2)** | Breast Reconstruction | 0 | 1 |
| | Wound Closure | 0 | 1 |

## Reporting of outcomes

The majority of subjective outcomes reported consisted of patient-reported outcome measures (PROMs), including visual analogue scale (VAS) pain scores and joint- and disease-specific functional outcome measures. A variety of validated and non-validated scores were used depending on the condition being studied. The vast majority (93%) of MSK studies used at least one PROM in their study (Table 4). Of these, 66% of them reported scores from PROMs favoring PRP over other treatment groups. In contrast, only 49% of studies from all other specialties used a PROM to assess outcomes (p<0.01).

**Table 2. Percentage of studies reporting details on composition of platelet-rich plasma used.**

| Specialty (Total Number of Studies) | Composition | Platelet Concentration | Leukocyte Concentration |
|---|---|---|---|
| **Cosmetic (19)** | 30% (5) | 100% (5/5) | 20% (1/5) |
| **Dermatology (3)** | 33% (1) | 100% (1/1) | 0% (0/1) |
| **Musculoskeletal (97)** | 37% (36) | 100% (36/36) | 56% (20/36) |
| **Neurology (3)** | 67% (2) | 100% (2/2) | 100% (2/2) |
| **Other (10)** | 0% (0) | 0% (0/0) | 0% (0/0) |
| **TOTAL (132)** | **34% (44)** | **100% (44/44)** | **52% (23/44)** |

**Table 3. Studies using an activator for platelet-rich plasma.**

| Specialty (Total Number of Studies) | % of Studies Using Activator | Calcium (# of Studies) | Thrombin (# of Studies) | Other (# of Studies) |
|---|---|---|---|---|
| Cardiothoracic Surgery (1) | 100% (1) | 0 | 1 | 0 |
| Cosmetic (19) | 74% (14) | 13 | 0 | 1 |
| Dermatology (3) | 67% (2) | 2 | 0 | 0 |
| Musculoskeletal (97) | 36% (35) | 26 | 5 | 4 |
| Neurology (3) | 0% (0) | | | |
| Ophthalmology (1) | 0% (0) | | | |
| Oral Maxillofacial Surgery (6) | 50% (3) | 2 | 0 | 1 |
| Plastic Surgery (2) | 100% (2) | 2 | 0 | 0 |

Objective outcomes assessed included imaging studies, examination-based quantification, and time to return to play. Among MSK studies, 57% reported at least one objective outcome measure, and 48% of these studies demonstrated favorability of PRP on an objective outcome measure over other treatments (Table 4). Among studies from all other specialties, 66% used an objective outcome measure (p = 0.43).

## Overall favorability of PRP compared to control treatment

Overall, 61% of the studies found PRP to be favorable over control treatment (Table 5). In 33 studies, PRP was administered as an adjunctive treatment and compared to primary treatment without PRP. These primary treatments ranged from other nonoperative therapies to surgery. Among studies utilizing PRP as adjunctive therapy, 66% found PRP to be favorable compared to controls. Among these studies comparing PRP to saline, 50% found PRP to be favorable compared to saline. With regards to specialty, 58% of MSK studies found PRP to be favorable over control, while 67% of studies from all other specialties found PRP to be favorable over control (p = 0.42). Among the few studies from dermatology, neurology, ophthalmology, and plastic surgery, 100% studies found PRP to be favorable over control.

## Risk of bias

Majority of studies were assessed using the Cochranes Risk of Bias Tool, 80% (n = 106). Among these studies, 30% (n = 32) were assessed to be "Low" risk of bias, 25% (n = 26) were found to have "Some Concerns", and 45% (n = 48) were assessed to be "High" risk of bias (S2 Appendix). The remaining 20% (n = 26) were assessed using MINORS with an average score of 17.4, SD = 2.96 (S3 Appendix).

## Discussion

The present systematic review found that the vast majority of published level I and II clinical studies investigating PRP therapy across all medical specialties have been conducted for MSK conditions. Among all studies analyzed, only 33% reported details on the composition of PRP

**Table 4. Percentage of studies reporting subjective and objective outcomes.**

| Specialty (Total Number of Studies) | Studies Reporting Subjective Outcomes | Studies Reporting Favorable Subjective Outcomes for PRP | Studies Reporting Objective Outcomes | Studies Reporting Favorable Objective Outcomes for PRP |
|---|---|---|---|---|
| Musculoskeletal (97) | 93% (90) | 66% (60/90) | 57% (56) | 48% (28/56) |
| Cosmetic (19) | 63% (12) | 58% (7/12) | 74% (14) | 64% (14/19) |
| Other (16) | 31% (5) | 60% (3/5) | 56% (9) | 78% (7/9) |

**Table 5. Percentage of studies reporting favorability of platelet-rich plasma over control treatment.**

| Specialty (Total Number of Studies) | Primary Treatment (PRP used as adjunct therapy) | Saline | Nonprocedural Treatment (e.g., physical therapy) | Hyaluronic Acid | Procedure | Other | TOTAL |
|---|---|---|---|---|---|---|---|
| **Musculoskeletal (97)** | 64% (14/22) | 44% (8/18) | 70% (14/20) | 53% (9/17) | 67% (4/6) | 57% (8/14) | **58% (57/97)** |
| **Cosmetic (19)** | 60% (3/5) | 71% (5/7) | | | | 33% (1/3) | 50% (2/4) | **58% (11/19)** |
| **Oral Maxillofacial Surgery (6)** | 100% (1/1) | 0% (0/1) | | 100% (2/2) | 50% (1/2) | | **67% (4/6)** |
| **Dermatology (3)** | 100% (3/3) | | | | | | **100% (3/3)** |
| **Neurology (3)** | | | 100% (2/2) | | | 100% (1/1) | **100% (3/3)** |
| **Plastic Surgery (2)** | 100% (1/1) | | | 100% (1/1) | | | **100% (2/2)** |
| **Ophthalmology (1)** | | | | | | 100% (1/1) | **100% (1/1)** |
| **Cardiothoracic Surgery (1)** | 0% (0/1) | | | | | | **0% (0/1)** |
| **TOTAL (132)** | **66% (22/33)** | **50% (13/26)** | **73% (16/22)** | **60% (12/20)** | **55% (6/11)** | **60% (12/20)** | **61% (81/132)** |

used, and only 17% reported the leukocyte component of PRP used. MSK studies were more likely to use PROMs as their primary outcome measure, while studies from other specialties were more likely to use objective outcomes. Overall, 61% of the studies found PRP to be favorable over control treatment, with no difference in favorable reporting between MSK and other medical specialties.

In recent years, there has been an explosion in the clinical practice of PRP treatment despite little evidence to support its use for most indications. Platelets, which contain various growth factors and cytokines that initiate and regulate healing, play a role in the normal mechanisms for tissue repair. As such, PRP, with a platelet concentration above the baseline of autologous blood, is thought to deliver more proteins, cytokines, and other bioactive factors to augment or promote the healing process. However, uncertainty remains as to the critical platelet concentration that correlates with appreciable augmentation of healing, which may vary depending on the mode of delivery, tissues involved, and pathology being treated. Moreover, positive clinical responses to PRP may be attributed to anti-inflammatory effects rather than a predominantly anabolic response as PRP therapy was originally intended to induce [1,20,21]. Improved understanding of the underlying structural and compositional deficiencies of the injured tissue, along with characterization of the specific components in PRP that are beneficial in the pathophysiology being treated, will help to define how or if PRP can be symptom-modifying and/or structure-modifying [1].

As a minimally manipulated tissue and autologous blood product, PRP has avoided the regulatory hurdles of extensive preclinical and clinical trial testing, and as a result, its clinical practice may always outpace the supporting scientific data. Rigorous, controlled, human clinical studies on PRP therapy are therefore crucial in the evaluation process, not only to evaluate its efficacy, but also to aid in defining the critical components within the PRP that are responsible for clinical improvement. Within orthopaedics, there has been numerous calls for minimal reporting standards in clinical studies regarding PRP preparation and composition in order to allow for comparison among studies and reproducibility [12–15]. However, among other medical specialties, similar calls for minimal reporting standards appear to be lacking. Even with the orthopaedic call for minimal reporting standards, this study found that only a third of

studies among all specialties, including MSK specialties, provided details on the PRP processing and characteristics, with only half of those studies performing leukocyte analysis. The clinical ramifications and cellular effects of leukocyte-rich versus leukocyte-poor PRP continue to be debated. Although some believe that leukocyte (neutrophil)-rich PRP is associated with pro-inflammatory effects due to an elevated level of catabolic cytokines which antagonize the anabolic factors contained within the platelets [22,23], others have reported the opposite [21]. The differences in the concentrations of bioactive molecules, including interleukin 1 receptor antagonist, platelet-derived growth factor, and matrix metalloproteinases, within leukocyte-rich PRP versus leukocyte-poor PRP make leukocyte reporting one of key elements in the clinical evaluation of PRP [21]. Furthermore, because the majority of commercially available PRP systems do not allow for user-titration of the leukocyte component, and because leukocyte concentration appears to important depending on the disease being treated [1,24], the lack of leukocyte reporting among current PRP studies only adds to this debate and makes it difficult to delineate the optimal PRP formulation to use. We propose that future PRP studies include the following PRP characterization details; concentration of platelets, activation status/activator used, leukocyte concentration, volume of therapeutic dose, and number of doses administered. The wide variability in the reporting of PRP processing, composition, activation, and delivery in the highest level of evidence clinical studies all contribute to the uncertainty and skepticism of PRP therapy within the medical community, and systematic standardization of reporting and classification systems, not just in orthopaedics but among all medical specialties, is a necessary step in translating PRP therapy into clinically meaningful treatment.

Other than the disparity in the number of published studies per specialty, this study found that MSK studies (93%) were more likely measure outcomes using PROMs compared to studies from other specialties (49%). Although some diseases may not be as amenable to subjective outcome measurement, in most scenarios, PRP is often being administered as a self-paying treatment option for cosmesis and quality of life. Therefore, PROMs to gauge satisfaction and other measures of quality of care from a patient's perspective should be a necessary, if not the primary, outcome in all PRP studies. Although PROMs exist in dermatology and aesthetic surgery [25–27], they may not be as widely used and validated compared to the myriad MSK PROMs [28]. In non-MSK studies, objective measures were more often the primary outcome measure assessed (e.g., maintenance of breast volume on imaging and clinical evaluation after PRP-supplemented breast reconstruction, quantitative assessment of hair growth and density after PRP injection for alopecia). However, these metrics don't capture the patient's perception of their health status, which clinicians should consider as the gold-standard outcome for evaluation.

There are several limitations to this review. Data on the reporting of other suggested criteria for PRP treatment evaluation, such as whole blood storage, whole blood characteristics, and patient usage of anti-inflammatory or anti-platelet medications, were not specifically elicited in this study. Given the wide variability or lack of reporting of PRP processing and composition, it is hypothesized that the reporting on whole blood details would be similarly scarce among all studies. Finally, this review of the literature would be subject to publication bias as 45% of the studies assessed using Cochrane were found to have "High" risk of bias. This adds to the call for standardization of not only composition of PRP but also the design and reporting of PRP studies.

In summary, the vast majority of level I and II clinical studies investigating PRP have been conducted for MSK injuries, with only a handful of studies conducted for conditions in other medical specialties. Studies that reported details on PRP processing and composition were in the minority, and PROMs were not often used as an outcome measure in non-MSK studies. Because the clinical practice of PRP may always outpace the supporting scientific data,

rigorous reporting in human clinical studies across all medical specialties is crucial for evaluating the effects of PRP and moving towards disease-specific and individualized treatment.

## Supporting information

**S1 Checklist. PRISMA PRP checklist.**
(PDF)

**S1 Appendix. Included studies.**
(DOCX)

**S2 Appendix. Cochrane risk of bias.**
(DOCX)

**S3 Appendix. MINORS.**
(DOCX)

## Acknowledgments

Investigation performed at the University of California, Irvine, Irvine, CA.

## Author Contributions

**Conceptualization:** Sarah Oyadomari, Dean Wang.

**Data curation:** Jaron Nazaroff, Nolan Brown.

**Formal analysis:** Jaron Nazaroff, Sarah Oyadomari, Nolan Brown, Dean Wang.

**Investigation:** Jaron Nazaroff, Sarah Oyadomari, Dean Wang.

**Methodology:** Jaron Nazaroff, Sarah Oyadomari, Dean Wang.

**Supervision:** Dean Wang.

**Validation:** Jaron Nazaroff, Dean Wang.

**Writing – original draft:** Jaron Nazaroff, Sarah Oyadomari, Nolan Brown, Dean Wang.

**Writing – review & editing:** Jaron Nazaroff, Dean Wang.

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
