## [Decision Letter · Decision Letter 0]

8 Jan 2021

PONE-D-20-22836

Reporting in Clinical Studies on Platelet-Rich Plasma Therapy Among All Medical Specialties: A Systematic Review of Level I and II Studies

PLOS ONE

Dear Dr. Wang,

Thank you for submitting your manuscript to PLOS ONE. After careful consideration, we feel that it has merit but does not fully meet PLOS ONE’s publication criteria as it currently stands. Therefore, we invite you to submit a revised version of the manuscript that addresses the points raised during the review process.

We look forward to receiving your revised manuscript.

Kind regards,

Ahmed Negida, MD

Academic Editor

PLOS ONE

Journal Requirements:

2. Please include captions for your Supporting Information files at the end of your manuscript, and update any in-text citations to match accordingly. Please see our Supporting Information guidelines for more information: http://journals.plos.org/plosone/s/supporting-information

Reviewers' comments:

Reviewer's Responses to Questions

**Comments to the Author**

1. Is the manuscript technically sound, and do the data support the conclusions?

Reviewer #1: Partly

Reviewer #2: Yes

Reviewer #3: Partly

Reviewer #4: Yes

Reviewer #5: Yes

2. Has the statistical analysis been performed appropriately and rigorously? 

Reviewer #1: Yes

Reviewer #2: Yes

Reviewer #3: Yes

Reviewer #4: Yes

Reviewer #5: I Don't Know

3. Have the authors made all data underlying the findings in their manuscript fully available?

Reviewer #1: Yes

Reviewer #2: Yes

Reviewer #3: Yes

Reviewer #4: Yes

Reviewer #5: Yes

4. Is the manuscript presented in an intelligible fashion and written in standard English?

Reviewer #1: Yes

Reviewer #2: Yes

Reviewer #3: Yes

Reviewer #4: Yes

Reviewer #5: Yes

5. Review Comments to the Author

Reviewer #1: This study reviewed the quantity of level I/II studies within 8 medical specialties which used PRP therapy. The data collection and statistical analysis were proper. The conclusions basically support the results. However, there are still some issues needed to be addressed:

1. The aim and the purpose of the study is not clearly stated. The indication, the pathological changes, the dose of PRP, etc. of various medical specialties are totally different. Simply comparing the PRP processing, characterization and overall outcomes among 8 medical specialties does not provide suggestions to clinical decision making.

2. In this study, the authors focused on the condition treated, PRP processing and characterization, delivery, control group, and assessed outcomes. But like the authors described in lines 121-123, the quality of PRP treatment, the total volume and the treatment window differed significantly from study to study. When there is such considerable difference, comparisons between MSK studies and other specialties seems less clinically meaningful.

3. Like the authors mentioned in DISCUSSIONS, there are still many controversies exist in the application of PRP treatment, including the concentration of leukocytes, the concentration of platelets, etc. If not under similar conditions of PRP therapies, the reviews of different medical specialties would not help the improvement of PRP preparation and utilization.

Reviewer #2: Good review, though many questions are yet to be answered in the reviewed studies. I hope this study will help future researchers in this topic to pay attention to these specific queries. Composition, techniques of usage and complications

Reviewer #3: GENERAL IMPRESSION:

This is an interesting paper that systematically reviewed and assessed Platelet-Rich Plasma (PRP) therapy among all medical specialties, The authors’ work provides a broad evaluation of level I/II studies across all the medical subspecialties combined, unlike most of the literature reviews on PRP, which usually covers a specific subspecialty (e.g. Dermatology, Cosmetics, Musculoskeletal ...etc.).

Further, the authors assessed all the included Level I/II studies in their review for bias and found that more than 40% have a high risk of bias, which I find as a notable and good contribution, and consolidation to the literature consensus that the use of PRP in clinical practice lacks solid evidence.

MAJOR COMMENTS:

1- I consider the author’s work on assessing the reviewed studies for the risk of bias using ‘’Cochrane Risk of Bias Tool’’ and ‘’MINORS’’ as the main contribution in the paper; other contributions as (reporting the favorability of PRP over control treatment), or (inconsistency in reporting of PRP composition), is not necessarily novel or surprising. Besides, the high risk of bias the authors found is a non-dramatic result, under the umbrella of ‘’no clear protocol for PRP’’ use and reporting.

2- In the limitations section, the authors mentioned that (platelet-rich fibrin) or (other platelet-rich formulations) may not have been captured by their search conditions. Yet, they did not provide a clear justification of why they decided not to include those studies in their work. When searched on PubMed for (platelet-rich fibrin) and added the filter of RCT, 157 results appeared. I believe the potential effect of such many studies -if were to be included- on the results could be notable.

3- The authors found that 30% of the reviewed papers had (low risk of bias) and 45% had (high risk of bias). I expected that the authors would provide a subgroup descriptive analysis and discussion, which could be a better demonstration of their work or even could provide a stronger judge. Like for example, if the subgroup analysis of the “low bias’’ studies showed an equivocal effect of PRP, that would be an absolutely good reference point.

4- It is agreed that there is a lack of solid evidence to support the use of PRP in clinical practice. But lack of uniformity in reporting doesn’t necessarily abolish the possibility that PRP could be a potentially effective treatment method. The authors in their discussion didn’t provide a good insight on how to particularly overcome this problem, while in fact, they should, so the readers would get a better informing review and minimize the risk of confirmation bias.

MINOR COMMENTS:

- Table 4 has flipped numerator and denominator in cell 3 in the last column on the right ( 64% (19/14) ).

Thanks.

AA

Reviewer #4: This is a good review of the clinical studies in Platelet-Rich Plasma (RPR) therapy among medical specialties, a field that has boomed in recent years. The article proposes an interesting perspective on the quantity of level I and II studies among medical specialties and the various levels of reporting in these studies. Interestingly, given the abundance of studies from orthopedic literature, the authors performed a clever comparison between the level of reporting between MSK studies and those from other medical fields. The review suggests a critical point of inconsistent reporting of RPR protocols, the variety of composition of RPR given, and the subsequent need for meticulous reporting of RPR protocols in human studies for proper evaluation of the efficacy of this therapy.

Major comments:

1. The Need for multifaceted search: The authors did a good job researching the revenant articles to include in the review. However, I suggest more detailed research such as manually reviewing reference lists of key articles, searching citations by using Web of Science, Google Scholar, and maybe experts’ group consultation to identify any other studies. Additionally, I feel that searching might need to be extended to published and unpublished reports in English and non-English Literature as well such as Japanese, Chinese, and WHO regional databases to avoid language bias. This might eventually decrease publication bias.

2. Unclear reproducibility of selection and assessment: The extraction of the relevant articles and risk of bias assessment needs to be performed by two independent blinded reviewers and a third reviewer in case of discrepancies to decrease the potential selection bias. The authors might also include the Kappa score to display the level of agreement between the reviewers.

Minor comments:

1. Page 3, conclusion: I suggest mentioning that “Knee osteoarthritis and tendinopathy being most commonly studied “after mentioning “The majority of level I/II clinical studies investigating PRP therapy across all 23 medical specialties have been conducted for MSK injuries “. This is to give the readers explicit truth in the conclusion about the most commonly studied conditions in that field.

1. Figure 1; PRISMA Flow chart: I believe authors need to mention the number of articles excluded for the corresponding reasons. The authors mention in the flowchart that full text-excluded (n=92): Level 3, level 4, letters, reviews, protocols, supplemented PRP, follow up study. I believe it is better to mention that “levels of evidence III or higher (n=77), lack of a PRP 98 experimental arm (n=12), and follow-up reporting of an already included study (n=3)”. This facilitates a comprehensive understanding of the exclusion causes by looking at the flow chart without the need to look back at the manuscript.

Overall, the manuscript has great potential and deals with an interesting and recently expanding topic.

Reviewer #5: In the present article the authors conducted a systemic review to answer two questions; First: the quantity of level I and II PRP therapy concerned studies. Second, to determine the level of reporting in these studies in regard to PRP processing, composition, activation, delivery, and outcome assessment.

After applying the inclusion and exclusion criteria of systemic review studies, 132 published studies were included in the systemic review. Level I and level II studies represent 71% and 29% respectively of the PRP studies. The authors reported that the vast majority of level I and level II studies had been conducted for MSK injuries, and 33% of the studies provided details on PRP processing or characteristics of PRP

Reviewer comment

I think the authors presented an adequate answers to questions raised in the abstract and in the introduction. In this article the author pointed to the lack in the in formations and data concerned the PRP composition, processing and characteristics. Furthermore, a great variety in reporting the PRP processing and characteristics among highest level of clinical studies was also reported by the authors, an issue that may contribute the uncertainty of PRP within the medical community. However, It is important to know whether the data extracted from the published articles were independently evaluated by the authors or not??. This is very important and should be clarified in the manuscript.

Reviewer comment

1. It is not clear what was exclusion criteria after screening the title screening ( line 96).

2. Table 1 : please indicate whether the number in the table represent number of studies or percentage.

3. The flowchart for the inclusion and exclusion criteria of studies included in the systemic review should be revised. For example, the follow-up study belong to level 3 study. Please report the exact number of level 3, and 4, letters, reviews, protocols and supplemented PRP which were excluded from the systemic review.

4. The discussion is very long

5. Statement in line 104 and 105 is not clear to the general reader, please explain or simplify “Of the MSK studies, 76% were level I evidence, and among all other studies, 57%

were level I evidence (p<0.05)”

6. Line 197-199, Please provide reference for this statement “Nevertheless, this study found that only a third of studies among all specialties, including MSK specialties, provided details on the PRP processing and characteristics, with only half of those studies performing leukocyte analysis”

6. PLOS authors have the option to publish the peer review history of their article (what does this mean?). If published, this will include your full peer review and any attached files.

Reviewer #1: No

Reviewer #2: **Yes: **Ahmed H. K. Abdelaal

Reviewer #3: No

Reviewer #4: No

Reviewer #5: No

---

## [Author Response · Author response to Decision Letter 0]

16 Feb 2021

Review Comments to the Author

Reviewer #1: This study reviewed the quantity of level I/II studies within 8 medical specialties which used PRP therapy. The data collection and statistical analysis were proper. The conclusions basically support the results. However, there are still some issues needed to be addressed:

1. The aim and the purpose of the study is not clearly stated. The indication, the pathological changes, the dose of PRP, etc. of various medical specialties are totally different. Simply comparing the PRP processing, characterization and overall outcomes among 8 medical specialties does not provide suggestions to clinical decision making.

Author response: The aim and the purpose of this study is stated in lines 53-58. The results of this study were not indicated to suggest clinical decision making but rather highlight the need for the highest level of evidence PRP studies that provide adequate reporting of PRP composition and trial design. 

2. In this study, the authors focused on the condition treated, PRP processing and characterization, delivery, control group, and assessed outcomes. But like the authors described in lines 121-123, the quality of PRP treatment, the total volume and the treatment window differed significantly from study to study. When there is such considerable difference, comparisons between MSK studies and other specialties seems less clinically meaningful.

Author response: We agree with this statement and have this included in our discussion (lines 221- 225).

3. Like the authors mentioned in DISCUSSIONS, there are still many controversies exist in the application of PRP treatment, including the concentration of leukocytes, the concentration of platelets, etc. If not under similar conditions of PRP therapies, the reviews of different medical specialties would not help the improvement of PRP preparation and utilization.

Author response: The purpose of this study was to present the existing highest level of evidence PRP studies among all medical specialties and evaluate the quality of reporting of PRP characterization in these studies. We agree with the reviewer’s assessment and it is our hope that the results of this study will highlight the need of more detailed reporting in such studies among the medical community. By comparing across specialties, we hope that the standards of reporting will be universal and not restricted within a single specialty. 

Reviewer #2: Good review, though many questions are yet to be answered in the reviewed studies. I hope this study will help future researchers in this topic to pay attention to these specific queries. Composition, techniques of usage and complications

Reviewer #3: GENERAL IMPRESSION:

This is an interesting paper that systematically reviewed and assessed Platelet-Rich Plasma (PRP) therapy among all medical specialties, The authors’ work provides a broad evaluation of level I/II studies across all the medical subspecialties combined, unlike most of the literature reviews on PRP, which usually covers a specific subspecialty (e.g. Dermatology, Cosmetics, Musculoskeletal ...etc.).

Further, the authors assessed all the included Level I/II studies in their review for bias and found that more than 40% have a high risk of bias, which I find as a notable and good contribution, and consolidation to the literature consensus that the use of PRP in clinical practice lacks solid evidence.

MAJOR COMMENTS:

1- I consider the author’s work on assessing the reviewed studies for the risk of bias using ‘’Cochrane Risk of Bias Tool’’ and ‘’MINORS’’ as the main contribution in the paper; other contributions as (reporting the favorability of PRP over control treatment), or (inconsistency in reporting of PRP composition), is not necessarily novel or surprising. Besides, the high risk of bias the authors found is a non-dramatic result, under the umbrella of ‘’no clear protocol for PRP’’ use and reporting.

Author Response: We agree that the results from the Cochrane’s and MINOR tools are a main factor in our overall conclusion that the standard of reporting for PRP clinical studies is lacking, in both composition and trial reporting/design. (line 240)

2- In the limitations section, the authors mentioned that (platelet-rich fibrin) or (other platelet-rich formulations) may not have been captured by their search conditions. Yet, they did not provide a clear justification of why they decided not to include those studies in their work. When searched on PubMed for (platelet-rich fibrin) and added the filter of RCT, 157 results appeared. I believe the potential effect of such many studies -if were to be included- on the results could be notable.

Author response: With the expanded use and development of PRP, there have been many modifications to the processing of autologous blood which is thought to yield a distinct product from PRP, such as platelet rich fibrin (PMID 3591032). This line has been deleted for clarification. 

3- The authors found that 30% of the reviewed papers had (low risk of bias) and 45% had (high risk of bias). I expected that the authors would provide a subgroup descriptive analysis and discussion, which could be a better demonstration of their work or even could provide a stronger judge. Like for example, if the subgroup analysis of the “low bias’’ studies showed an equivocal effect of PRP, that would be an absolutely good reference point.

Author response: This would be interesting data. However, we feel that the current inconsistent reporting of PRP and the heterogeneity of methods used to evaluate its effects (characterization, use of PROMs vs objective data, control type used, ect.), make the low vs high bias comparison difficult to perform across medical specialities

4- It is agreed that there is a lack of solid evidence to support the use of PRP in clinical practice. But lack of uniformity in reporting doesn’t necessarily abolish the possibility that PRP could be a potentially effective treatment method. The authors in their discussion didn’t provide a good insight on how to particularly overcome this problem, while in fact, they should, so the readers would get a better informing review and minimize the risk of confirmation bias.

Author response: Added to discussion, line 210-213.

MINOR COMMENTS:

- Table 4 has flipped numerator and denominator in cell 3 in the last column on the right ( 64% (19/14) ).

Author response: Thank you. We have made that correction. 

Thanks.

AA

Reviewer #4: This is a good review of the clinical studies in Platelet-Rich Plasma (RPR) therapy among medical specialties, a field that has boomed in recent years. The article proposes an interesting perspective on the quantity of level I and II studies among medical specialties and the various levels of reporting in these studies. Interestingly, given the abundance of studies from orthopedic literature, the authors performed a clever comparison between the level of reporting between MSK studies and those from other medical fields. The review suggests a critical point of inconsistent reporting of RPR protocols, the variety of composition of RPR given, and the subsequent need for meticulous reporting of RPR protocols in human studies for proper evaluation of the efficacy of this therapy.

Major comments:

1. The Need for multifaceted search: The authors did a good job researching the revenant articles to include in the review. However, I suggest more detailed research such as manually reviewing reference lists of key articles, searching citations by using Web of Science, Google Scholar, and maybe experts’ group consultation to identify any other studies. Additionally, I feel that searching might need to be extended to published and unpublished reports in English and non-English Literature as well such as Japanese, Chinese, and WHO regional databases to avoid language bias. This might eventually decrease publication bias.

Author Response: Thank you for your comment. We did manually review the reference list of key articles and added this detail to the methods. We believe that searching 3 major databases (medline, Cochrane, and EMBASE) for English language articles is very comprehensive and the standard for systematic reviews (Harris et al 2013). Including non-English articles would not be realistic as we would not be able to understand the articles and analyze them for this systematic review. 

2. Unclear reproducibility of selection and assessment: The extraction of the relevant articles and risk of bias assessment needs to be performed by two independent blinded reviewers and a third reviewer in case of discrepancies to decrease the potential selection bias. The authors might also include the Kappa score to display the level of agreement between the reviewers.

Author response: We had two reviewers perform the literature search and risk of bias assessment (line 72). The final articles and risk of bias assessment scores were then reviewed by the senior author (line 92). 

Minor comments:

1. Page 3, conclusion: I suggest mentioning that “Knee osteoarthritis and tendinopathy being most commonly studied “after mentioning “The majority of level I/II clinical studies investigating PRP therapy across all 23 medical specialties have been conducted for MSK injuries “. This is to give the readers explicit truth in the conclusion about the most commonly studied conditions in that field.

Author response: Added to line 23.

1. Figure 1; PRISMA Flow chart: I believe authors need to mention the number of articles excluded for the corresponding reasons. The authors mention in the flowchart that full text-excluded (n=92): Level 3, level 4, letters, reviews, protocols, supplemented PRP, follow up study. I believe it is better to mention that “levels of evidence III or higher (n=77), lack of a PRP 98 experimental arm (n=12), and follow-up reporting of an already included study (n=3)”. This facilitates a comprehensive understanding of the exclusion causes by looking at the flow chart without the need to look back at the manuscript.

Author response: PRISMA diagram updated to include proposed additions. 

Overall, the manuscript has great potential and deals with an interesting and recently expanding topic.

Reviewer #5: In the present article the authors conducted a systemic review to answer two questions; First: the quantity of level I and II PRP therapy concerned studies. Second, to determine the level of reporting in these studies in regard to PRP processing, composition, activation, delivery, and outcome assessment.

After applying the inclusion and exclusion criteria of systemic review studies, 132 published studies were included in the systemic review. Level I and level II studies represent 71% and 29% respectively of the PRP studies. The authors reported that the vast majority of level I and level II studies had been conducted for MSK injuries, and 33% of the studies provided details on PRP processing or characteristics of PRP

Reviewer comment

I think the authors presented an adequate answers to questions raised in the abstract and in the introduction. In this article the author pointed to the lack in the in formations and data concerned the PRP composition, processing and characteristics. Furthermore, a great variety in reporting the PRP processing and characteristics among highest level of clinical studies was also reported by the authors, an issue that may contribute the uncertainty of PRP within the medical community. However, It is important to know whether the data extracted from the published articles were independently evaluated by the authors or not??. This is very important and should be clarified in the manuscript.

Author response: We had two authors independently evaluate the extracted data (Line 85). 

Reviewer comment

1. It is not clear what was exclusion criteria after screening the title screening ( line 96).

Author response: Exclusion criteria is mentioned on line 68. 

2. Table 1 : please indicate whether the number in the table represent number of studies or percentage.

Author response: Indicated in table description. 

3. The flowchart for the inclusion and exclusion criteria of studies included in the systemic review should be revised. For example, the follow-up study belong to level 3 study. Please report the exact number of level 3, and 4, letters, reviews, protocols and supplemented PRP which were excluded from the systemic review.

Author response: Flow chart has been updated. 

4. The discussion is very long

5. Statement in line 104 and 105 is not clear to the general reader, please explain or simplify “Of the MSK studies, 76% were level I evidence, and among all other studies, 57%

were level I evidence (p<0.05)”

Author response: Sentence revised (line 110). 

6. Line 197-199, Please provide reference for this statement “Nevertheless, this study found that only a third of studies among all specialties, including MSK specialties, provided details on the PRP processing and characteristics, with only half of those studies performing leukocyte analysis”

Author response: Provided reference (line 203).

---

## [Decision Letter · Decision Letter 1]

30 Mar 2021

Reporting in Clinical Studies on Platelet-Rich Plasma Therapy Among All Medical Specialties: A Systematic Review of Level I and II Studies

PONE-D-20-22836R1

Dear Dr. Wang,

We’re pleased to inform you that your manuscript has been judged scientifically suitable for publication and will be formally accepted for publication once it meets all outstanding technical requirements.

Kind regards,

Ahmed Negida, MD

Academic Editor

PLOS ONE

Additional Editor Comments (optional):

Reviewers' comments:

Reviewer's Responses to Questions

**Comments to the Author**

1. If the authors have adequately addressed your comments raised in a previous round of review and you feel that this manuscript is now acceptable for publication, you may indicate that here to bypass the “Comments to the Author” section, enter your conflict of interest statement in the “Confidential to Editor” section, and submit your "Accept" recommendation.

Reviewer #1: All comments have been addressed

Reviewer #2: All comments have been addressed

Reviewer #3: (No Response)

Reviewer #4: All comments have been addressed

2. Is the manuscript technically sound, and do the data support the conclusions?

Reviewer #1: Yes

Reviewer #2: Yes

Reviewer #3: Yes

Reviewer #4: Yes

3. Has the statistical analysis been performed appropriately and rigorously? 

Reviewer #1: Yes

Reviewer #2: Yes

Reviewer #3: Yes

Reviewer #4: Yes

4. Have the authors made all data underlying the findings in their manuscript fully available?

Reviewer #1: Yes

Reviewer #2: Yes

Reviewer #3: Yes

Reviewer #4: Yes

5. Is the manuscript presented in an intelligible fashion and written in standard English?

Reviewer #1: Yes

Reviewer #2: Yes

Reviewer #3: Yes

Reviewer #4: Yes

6. Review Comments to the Author

Reviewer #1: (No Response)

Reviewer #2: Again, great job and perfect effort was done by the authors which highlighted such an important practical and debatable issue in our MSK prctice

Reviewer #3: Reviewer 3:

MAJOR COMMENTS:

1- Author Response: We agree that the results from the Cochrane’s and MINOR tools are a main factor in

our overall conclusion that the standard of reporting for PRP clinical studies is lacking, in both

composition and trial reporting/design. (line 240)

Review 3 comment : Thank you.

2-Author response: With the expanded use and development of PRP, there have been many modifications

to the processing of autologous blood which is thought to yield a distinct product from PRP, such as

platelet rich fibrin (PMID 3591032). This line has been deleted for clarification.

Review3 comment: Thank you.

3- Author response: This would be interesting data. However, we feel that the current inconsistent

reporting of PRP and the heterogeneity of methods used to evaluate its effects (characterization, use of

PROMs vs objective data, control type used, ect.), make the low vs high bias comparison difficult to

perform across medical specialties

Reviewer3 comment: I would suggest adding a column to table (5) to show [Percentage of studies with high risk bias] right to the column [Total].

4- Author response: Added to discussion, line 210-213.

Reviewer 3 comment: [210-213] is not a new added paragraph, it can be found in the original draft [203-206].

I believe the author meant to refer to a different paragraph. If not, I don't believe that paragraph addresses my comment.

Minor comments:

Author response: Thank you. We have made that correction.

Review3 comment: Thank you.

Reviewer #4: Thank you for submitting the revised version of the manuscript "Reporting in Clinical Studies on Platelet-Rich Plasma Therapy Among All Medical Specialties: A Systematic Review of Level I and II Studies. I do believe that the authors addressed most of my raised concerns and the manuscript now sounds more appropriate.

7. PLOS authors have the option to publish the peer review history of their article (what does this mean?). If published, this will include your full peer review and any attached files.

Reviewer #1: **Yes: **Xiaoxi Ji

Reviewer #2: No

Reviewer #3: No

Reviewer #4: No

---

## [Editor Report · Acceptance letter]

12 Apr 2021

PONE-D-20-22836R1 

Reporting in Clinical Studies on Platelet-Rich Plasma Therapy Among All Medical Specialties: A Systematic Review of Level I and II Studies 

Dear Dr. Wang:

I'm pleased to inform you that your manuscript has been deemed suitable for publication in PLOS ONE. Congratulations! Your manuscript is now with our production department. 

Kind regards, 

on behalf of

Dr. Ahmed Negida 

Academic Editor

PLOS ONE